# Community Detection in Semantic Networks: A Multi-View Approach

**DOI:** 10.3390/e24081141

**Published:** 2022-08-17

**Authors:** Hailu Yang, Qian Liu, Jin Zhang, Xiaoyu Ding, Chen Chen, Lili Wang

**Affiliations:** 1School of Computer Science and Technology, Harbin University of Science and Technology, Harbin 150001, China; 2School of Automatic Control Engineering, Harbin Institute of Petroleum, Harbin 150028, China; 3School of Computer Science and Technology, Chongqing University of Posts and Telecommunications, Chongqing 400065, China

**Keywords:** semantic social network, community detection, multi-view clustering, adaptive loss function, semantic information processing

## Abstract

The semantic social network is a complex system composed of nodes, links, and documents. Traditional semantic social network community detection algorithms only analyze network data from a single view, and there is no effective representation of semantic features at diverse levels of granularity. This paper proposes a multi-view integration method for community detection in semantic social network. We develop a data feature matrix based on node similarity and extract semantic features from the views of word frequency, keyword, and topic, respectively. To maximize the mutual information of each view, we use the robustness of L21-norm and F-norm to construct an adaptive loss function. On this foundation, we construct an optimization expression to generate the unified graph matrix and output the community structure with multiple views. Experiments on real social networks and benchmark datasets reveal that in semantic information analysis, multi-view is considerably better than single-view, and the performance of multi-view community detection outperforms traditional methods and multi-view clustering algorithms.

## 1. Introduction

With the rapid expansion of the network, the interaction of online users has increased greatly; people expand their social life in an unprecedented way. There are not only online social networks with millions of participants, such as Facebook, Twitter, and QQ, but also various offline social networks in every corner of our life. When BBS, news websites, or blogs are used to share views and deliver messages, a social network with similar interests and hobbies is formed. These social networks contain interesting patterns and attributes that have significant study value for evaluating people’s social behavior [1,2]. An important goal of social network analysis is to reveal the self-organization phenomenon behind the network topology, which can be achieved by identifying the community structure with highly connected nodes [3,4].

Community detection is one of the hotspots in complex network research. Its purpose is to find subgraphs with dense internal but sparse external connections [5]. Roughly speaking, community detection is to divide actors with social relationships into close and highly related groups [6]. Real social networks usually consist of multiple views, for example, the same news can be reported by different news organizations, the pictures shared on the website can have different text descriptions, and the meteorological data can be collected from different sensors. To make full use of multi-view information and improve clustering performance, multi-view clustering has been proposed and has begun to attract more and more attention. Multi-view clustering can fuse the complementary information hidden in each view [7,8,9], which provides high-quality implementation schemes for community detection.

Traditional community detection algorithm divides the topology of the network. Hierarchical clustering algorithm detects communities based on the similarity or the connection strength between nodes; commonly used algorithms include Newman fast algorithm [10], Newman greedy algorithm [11], and spectrum-based aggregation algorithm [12]. Spectral clustering algorithm [13,14] finds communities by analyzing the eigenvalues and eigenvectors of the Laplace matrix or standard matrix formed by the adjacency matrix. The modularity optimization algorithm detects communities in the network by optimizing the modularity function. Simulated annealing algorithm [15] and Louvain algorithm [16] are two popular algorithms. The improved modularity optimization algorithm [17,18] adopts the improved modularity function to different types of networks to realize community detection.

Although the above methods have achieved great success in the field of community detection, the majority of them are only effective for single view networks. Even if all views are integrated into a single view for community detection, it is difficult to increase performance because each view has its own properties. Using multi-view clustering for community detection, on the other hand, can take into account the diversity and complementarity of different views, effectively improving the completeness of the community structure.

The key to multi-view clustering is learning how to leverage the multiple attributes that are embedded in the object to divide it into different clusters. Early multi-view clustering integrated multi-view elements into traditional clustering methods, including multi-type reinforced clustering [19], dual view clustering based on EM and aggregation algorithm [20], multi-view clustering based on DBSCAN [21], etc. Multi-view clustering, in recent years, has primarily focused on developing clustering algorithms that conform to data characteristics for specific fields, such as collaborative training [22,23], multi-kernel learning [24,25], and multi-view graph clustering [26] for image and text data, and subspace clustering for matrix data [27,28]. These works demonstrate that multi-view clustering can detect common underlying structures shared by multiple perspectives and generate clusters by fusing views. However, no study has been conducted on community detection using different views integrated in semantic networks. Meanwhile, the efficiency of multi-view clustering for community detection has not been thoroughly tested. In particular, real-world semantic networks contain solely user attributes that do not directly possess a network form, which complicates and challenges community detection with multi-views.

To address these issues, this work presents a community detection method based on multi-view clustering. First, we reconstruct the semantic network to convert the user’s multi-attributes into network form; second, we apply the adaptive loss function to address the problem that the L1-norm and L2-norm are sensitive to bigger and smaller outliers, respectively. Finally, the data matrix of multiple perspectives is fused to generate communities. The main contributions of this paper include:(1)We propose to extract features of the network from multiple perspectives for community detection. The approach efficiently utilizes semantic information in social networks at various granularities. Compared with single-view community detection, multi-view community detection has better performance in modularity, accuracy, and F-score.(2)We propose an approach for reconstructing social networks. The approach utilizes a data matrix to describe the connections between user attributes in each perspective, which can subsequently be utilized to capture the intrinsic correlations across multiple views using matrix fusion. On the other hand, the method can avoid errors caused by the absence of data and relationships.(3)We present a multi-view community detection method based on the adaptive loss function. The method can decrease the impact of outlier points on community segmentation. Experiments show that the method is not only applicable to real social networks, but also outperforms traditional community detection methods when coping with other types of data.

## 2. Semantic Feature Representation of Nodes

The social network is composed of rich semantic information and complex semantic content. We define it as G=(P,O,D), where *P* is the node set, representing users in the social network; *O* is the edge set, representing the link relationship between social network users; *D* is the semantic information, which represents the document published by users. To capture the semantic components of user text from multiple views, we use word frequency, keyword, and topic as three perspectives of semantic features according to the semantic granularity from high to low, and represent the semantic features in the form of a data feature matrix.

### 2.1. Word Frequency

The lowest granularity representation of text information is word frequency. Word frequency analysis can objectively interpret abstract text data and detect implicit hot spots in the text according to the frequency of phrases. It is common in computer science, communication, and information science [29]. During COVID-19, for example, word frequency was widely used in pneumonia data analysis [30] and Twitter post analysis [31]. This paper extracts word frequency features from the text submitted by social network members and creates word frequency vectors. The word frequency is expressed by $f_{i,j}$; that is, the number of times the word $w_{i}$ appears in the document $d_{i}$, where d∈D.

First, the semantic information of the social network is pre-processed, which includes filtering and word segmentation. The processed semantic information is used to create corpus $D^{\prime }$, which is subsequently used to calculate the value of $f_{i}$. For example, if $w_{i}$ appears once in $D^{\prime }$, $f_{i}=1$; $w_{i}$ appears *n* times in $D^{\prime }$, $f_{i}=n$. Following that, the words are sorted in descending order of occurrence, and the number of features that compose the data feature matrix *X* is chosen based on the sorting results (details are described in Section 5). Finally, count the number of times these words appear in each text to create $f_{i,j}$. In matrix *X*, $x_{i,j}=f_{i,j}$. For example, if $w_{1}$ appears five times in $d_{1}$, $f_{1,1}=5$, the value of the element in the first row and first column of matrix *X* is 5, that is $x_{1,1}=5$.

In expressing semantic information, the word frequency perspective is insufficiently concise. It can retain the majority of the information in social texts, but it will also raise the probability of invalid information.

### 2.2. Keywords

Keywords are meaningful and representative words in documents that accurately describe the text content [32]. Compared to word frequency, keywords evaluate the structure and syntax of text information, which can eliminate text noise and reduce the number of semantic features. In this paper, the TF-IDF (term frequency-inverse document frequency) method is used as the measuring standard of keyword, and its value is used as the eigenvector for each word in the text. The formula of TF-IDF is as follows:(1)$$\begin{array}{c} TI=\frac{f_{i,j}}{\sum_{i}^{\left|w^{j}\right|}f_{i,j}}\times \log\frac{\left|D\right|}{\left|\left\{j:w_{i}\in d_{j}\right\}\right|} \end{array}$$
where $\left|w^{j}\right|$ represents the number of words in $d_{j}$, |D| represents the total number of texts in the corpus, i.e., the total number of semantic information released by social network users, and $\left|\left\{j:w_{i}\in d_{j}\right\}\right|$ represents the number of documents containing word $w_{i}$.

If the TF-IDF value of each word in the text is calculated directly, the data feature matrix *X* of the keyword perspective will be enormous. To tackle this problem, we define the parameter *t* to limit the number of matrix features (the choice of *t* is given in Section 5). After filtering, word segmentation, and part of speech tagging on the text in corpus $D^{\prime }$, we use Equation (Equation 1) to calculate the weight of each word relative to the corpus, and the words with top-*t* weight are used to create the keyword set kw. Equation (Equation 1) is used again to calculate the TF-IDF value of the keyword $kw_{i}$ in document $d_{j}$, which is marked as $TI_{i,j}$, and the data feature matrix *X* is filled with $x_{i,j}=TI_{i,j}$.

### 2.3. Topic

The topic is the condensation of the text, which has the highest level of granularity. In this paper, we extract topics in texts based on the LDA (Latent Dirichlet Allocation) model and construct the feature data matrix of the topic perspective.

The LDA model is an effective method to extract latent semantic information from text corpus. It is a three-layer Bayesian probability model, which is used to generate document topics, including words, topics, and documents. LDA models the document as a mixture of potential topics, and each topic can be further presented with a set of words. Therefore, in LDA, documents intuitively show multiple topics. After text preprocessing, each document is regarded as a mixture of topics in the corpus. Topics are composed of fixed words, and these topics are generated from the document collection. For example, the probability of science and technology topics has words: “chip” and “5g”, and the probability of entertainment topics has words: “Star” and “film”. Then, the document set has a probability distribution on the topic, where each word is considered to belong to one of the above topics. Through the probability distribution of the document on each topic, we can know the correlation between the document and each topic.

Therefore, the following steps can be used to describe the process of LDA generating documents. Firstly, it is assumed that the prior distribution of the semantic information of the social network is Dirichlet distribution; that is, for the text information d∈D published by any user, there is the topic distribution of the document $\theta_{d}=Dirichlet\left(\alpha \right)$. Then, assume that the prior distribution of the topic words is the Dirichlet distribution; that is, for any topic t∈T, there is the word distribution $\beta_{t}=Dirichlet\left(\eta \right)$. Next, for the *n*-th word in any semantic information $d_{j}$, we can get its topic number $Z_{d_{j},n}=\mathrm{multi}\left(\theta_{d_{j}}\right)$ from the theme distribution $\theta_{d}$. Finally, the probability distribution $w_{d_{j},n}=\mathrm{multi}\left(\beta_{d_{j}}\right)$ of the word $w_{d_{j},n}$ is known from the topic number $Z_{d_{j},n}$. In the above process, parameters α and η are hyper-parameter vectors, which determine the distribution of topics in the document and the distribution of words in the topics, respectively. The LDA generation process corresponds to the following joint distribution:(2)$$\begin{array}{cc} \; & P\left(\beta_{1:T},\theta_{1:D},Z_{1:D},w_{1:D}\right)\\ \; & =\prod_{i=1}^{T}P\left(\beta_{i}\right)\prod_{d=1}^{D}P\left(\theta_{d}\right)\left(\prod_{n=1}^{N}p\left(Z_{d,n}|\theta_{d}\right)p\left(w_{d,n}|\beta_{1:T},Z_{d,n}\right)\right) \end{array}$$

Partial dependencies are specified in Equation (Equation 2). Topic $Z_{d,n}$ depends on the topic distribution $\theta_{d}$ of the text information published by the user; word $w_{d,n}$ depends on the word distribution $\beta_{1:T}$ and topic $Z_{d,n}$ [33].

With the LDA model, the representation of the semantic features of social networks can be completed from the perspective of topics through the following processes. First, the most basic operations are carried out to clean and filter the semantic information. Then, determine the number of topics *T* (the method is given in Section 5), extract the topics from the information text by using the above steps of generating documents by LDA, and obtain the topic distribution $\theta_{d}$ of each information (including the weight of the topic to which the document belongs). Finally, the topic distribution $\theta_{d}$ and user published information are taken as the rows and columns of the data matrix, and the value of the topic distribution is used as the value of the data matrix to complete the semantic feature representation from the perspective of the topic.

The feature representation process of social networks based on word frequency, keywords, and topics are shown in Figure 1. The process includes: (1) Obtain the semantic information published by users from social networks to form a corpus. (2) Preprocessing the whole text set, including filtering meaningless words such as exclamation, preposition, and auxiliary word. (3) Feature extraction of the processed social network semantic information from the three perspectives described above. (4) The extracted features are transformed into vector representation and stacked together to form the feature data matrix of the social network.

## 3. Reconstruction of Social Networks

### 3.1. Node Representation

It can be seen from the previous section that the semantic information of social networks will be represented in the form of data matrix *X* from three angles. The storage structure of the data is shown in Figure 2. In the figure, the row of the matrix represents the value of the attribute, the column represents the node vector, the shaded part represents the value, and the blank part represents zero. The semantic information of each view in the social network will be represented by a data matrix *X*. Taking the keyword perspective as an example, suppose that the social network is composed of *n* users, the posts published by users represent semantic information *d*, the number of keywords *L* is the number of attributes, then the data matrix $X\in {}^{L\times n}$, and the value in the matrix is naturally the TF-IDF value of keyword kw in *d*. The node feature representation of each angle of the social network will be stored in the data matrix shown in Figure 2. Then, calculate the similarity between the node vectors to establish contact for users to complete the reconstruction of the social network.

Before introducing the following, we first give the symbolic representation used in this part, as shown in Table 1.

$X\in R^{\dim\times n}$ represents the data matrix of the social network, where dim represents the number of attributes of semantic information features in the social network, *n* represents the number of data (the number of users in the social network), $X^{v}$ represents the data matrix from the *v*-th perspective, and its *j*-th column vector represents $x_{j}^{v}\in R^{\dim\times 1}$, the ij-th element is represented as $x_{i,j}^{v}$. The trace and F-norm of matrix *X* can be expressed as Tr(X) and ${∥X∥}_{F}$. The p-norm of vector *x* is expressed as ${∥x∥}_{p}$.

### 3.2. Node Similarity Calculation

The connected matrix is generated after obtaining the data matrix of each view by computing the similarity between vectors; that is, to establish contact for users with similar semantic information. The correlation degree between semantic information can be measured by many statistical values, such as the most common cosine similarity calculation method, Pearson correlation coefficient used in the absence of dimension, and distance-based Gaussian kernel similarity calculation method. The first two methods rely on the defined measurement rules and ignore the local geometry of the data and the size of the vector itself. Gaussian kernel similarity is a measurement method that is based on distance, which is sensitive to noise and outliers in the data. Therefore, a data similarity matrix learning method based on sparse representation proposed by Nie et al. [34] is used in this paper. Compared with the above three similarity calculation methods, this method is more in line with the construction of the connected matrix of the social network, so that the users with a high degree of association in the social network (the feature vector distance corresponding to the semantic information published by the user) is small. Corresponding to a large similarity value, the similarity value between users with small correlation degree is small or even zero, and the sparse representation is robust to noise and outliers in the data [35]. The connected matrix can be obtained by solving the following problems:(3)$$\begin{array}{cc} \; & \min_{c_{i,j}}c_{i,j}{∥x_{i}-x_{j}∥}_{2}^{2}+\alpha \sum_{j}^{n}c_{i,j}^{2}\\ \; & s.t.\;{∥c_{i}∥}_{1}=1,c_{i,i}=0,c_{i,j}\geq 0 \end{array}$$

Here, α is a sparse factor. The following results can be obtained after calculation and derivation.
(4)$$\begin{array}{c} {\hat{c}}_{i,j}=\left\{\begin{array}{cc} \frac{a_{i,m+1}-a_{i,j}}{ma_{i,m+1}-\sum_{h=1}^{m}a_{i,h}} & j\leq m\\ 0 & j>m \end{array}\right. \end{array}$$
where $a_{i,j}={∥x_{i}-x_{j}∥}_{2}^{2}$ ,and will sort it from small to large, so that the learning of $c_{i}$ meets ${\hat{c}}_{i,m}>0$ and ${\hat{c}}_{i,m+1}=0$. In this paper, the matrix calculated by Equation (Equation 4) is called the connected matrix *C* of a single perspective of the social network. According to the connected matrix, the connection relationship between users can be known. Therefore, C is also an adjacency matrix. The matrix *C* can be used to obtain the incidence graph of social networks from a single perspective. Compared with fixed connection graph structures, such as the full connection graph and *k*-nearest neighbor graph, the above method can adapt the number of neighbors *m* of users. Compared with cosine similarity, Pearson correlation coefficient, and other methods, the connected matrix constructed in this way will have higher quality, can make up for the disadvantage that spectral clustering requires higher node similarity, and makes the effect of subsequent community discovery better.

## 4. Community Detection

Traditional community detection methods are usually used to deal with social networks from a single view, which will be weak when dealing with social networks from the multi-view. Therefore, this paper proposes a multi-view community detection method based on adaptive loss function (ALMV) to realize the community detection of multi-view social networks. Using the adaptive loss function, this method not only adapts the weights of each view, but also learns to get the final matrix after the fusion of multiple perspectives, which contains *k* connected components and can directly output the results of community detection. In this paper, this matrix is called the consensus matrix $S\in R^{n\times n}$. The approach will be described in the next section.

### 4.1. Adaptive Loss Function

Loss functions are usually constructed using l1-norm and l2-norm. For any vector *x*, l1-norm and l2-norm are defined as ${∥x∥}_{1}=\sum_{i}^{n}\left|x_{i}\right|$ and ${∥x∥}_{2}^{2}=\sum_{i}^{n}x_{i}^{2}$, respectively. Defining the loss function l1-norm is insensitive to larger outliers, but sensitive to smaller outliers. l2-norm is the opposite, which has a large impact on model learning. The adaptive loss function [36] can well neutralize the above problems. The specific definition of the function is as follows:(5)$$\begin{array}{c} {∥x∥}_{\sigma }=\sum_{i}^{n}\frac{\left(1+\sigma \right)x_{i}^{2}}{x_{i}^{2}+\sigma } \end{array}$$

Here, σ is an adaptive parameter. If vector *x* is extended to matrix *X*, it is equivalent to the neutralization of l21-norm and F-norm of the matrix, which are defined as ${∥x∥}_{2,1}=\sum_{i}^{n}{∥x_{i}∥}_{2}$ and ${∥x∥}_{F}^{2}=\sum_{i}^{n}{∥x_{i}∥}_{2}^{2}$, respectively. The adaptive loss function of the matrix is generalized as follows:(6)$$\begin{array}{c} {∥X∥}_{\sigma }=\sum_{i}^{n}\frac{\left(1+\sigma \right){∥x_{i}∥}_{2}^{2}}{{∥x_{i}∥}_{2}+\sigma } \end{array}$$

It can be seen from Equation (Equation 6) that the adaptive loss function is defined between L21-norm and F-norm. Therefore, both large outliers and small outliers can make use of the robustness of L21-norm and F-norm. In addition, it is easy to verify that ${∥X∥}_{\sigma }$ is nonnegative, convex, and quadratic differentiable, so it is ideal for the loss function and optimization function. When σ→0, ${∥X∥}_{\sigma }\to {∥X∥}_{2,1}$, and σ→∞, ${∥X∥}_{\sigma }\to {∥X∥}_{F}^{2}$. Therefore, a different σ can be selected according to different situations. This paper will use the adaptive loss function to learn the consensus matrix *S* of multi-view social networks to construct a consensus graph. Its implement will be introduced in the next section.

### 4.2. Multi-View Community Detection Based on Adaptive Loss Function

The connected matrix $C^{\left(v\right)}$ of each perspective of the social network can be reconstructed through Equation (Equation 4) since each $C^{\left(v\right)}$ will affect the resulting consensus graph matrix *S*, and the closer to *S*, the larger the weight ω will be assigned to the connected matrix $C^{\left(v\right)}$ from a single perspective; otherwise the smaller ω will be assigned. Therefore, this paper will learn consensus matrix *S* by automatically weighting the connected matrix from each view based on the adaptive loss function, which presents the following objective functions:(7)$$\begin{array}{cc} \; & \min_{S}\sum_{v=1}^{V}\left\{\omega_{v}{∥C^{\left(v\right)}-S∥}_{\sigma }\right\}\\ \; & s.t.\;1^{T}s_{i}=1,s_{i,j}\geq 0,\mathrm{rank}\left(L\right)=n-k \end{array}$$
where $s_{i}\in R^{n\times 1}$ is the *i*-th column of the consensus matrix *S*. $s_{i,j}$ is the *j*-th element of the column vector $s_{i}$. $\omega =\{\omega_{1},\ldots ,\omega_{v}\}$ is the weight of the connected matrix for each view in the social network. *L* is the Laplace matrix of *S*, L=R−B. *R* is the diagonal matrix, $r_{ii}=\sum_{j=1}^{n}s_{i,j}$, $B=(S^{T}+S)/2$. rank(L)=n−k is the rank constraint introduced to the Laplace matrix *L* of *S*, which gives *S* have *k* connected components, thus directly outputting the *k* community structures of the social network.

However, *L* depends on the target variable *S* and the rank constraint is non-linear, which makes Equation (Equation 7) difficult to optimize. Let $\lambda {}_{i}\left(L\right)$ represent the *i*-th smallest eigenvalue of *L*. Since *L* is a symmetric positive semidefinite matrix, λ(L) is a real number and non-negative [37]. Therefore, it can be seen that the eigenvalue of *L* satisfies $\lambda {}_{i}\left(L\right)\geq 0$ and $\sum_{i=1}^{k}\lambda_{i}\left(L\right)=0$. The rank constraint is also achieved, so Equation (Equation 7) can be expressed as follows:(8)$$\begin{array}{cc} \; & \min_{S}\sum_{v=1}^{V}\left\{\omega_{v}{∥C^{\left(v\right)}-S∥}_{\sigma }\right\}+\gamma \sum_{i=1}^{k}\lambda_{i}\left(L\right)\\ \; & s.t.\;1^{T}s_{i}=1,s_{i,j}\geq 0 \end{array}$$
where γ is the balance factor, which can increase or decrease its value accordingly when the connected component of the consensus matrix is greater than or less than *k*, until there exist *k* connected components. Then, according to the research of Fan [38], the following theorem exists:(9)$$\begin{array}{cc} \; & \sum_{i=1}^{k}\lambda_{i}\left(L\right)=\min_{F}\mathrm{Tr}\left(F^{T}LF\right)\\ \; & s.t.\;F^{T}F=I \end{array}$$
where $F\in R^{n\times k}$, and $F=\left\{f_{1},f_{2},\ldots ,f_{k}\right\}$ is composed of the eigenvector *f* corresponding to the smallest *k* eigenvalues. According to Equations (Equation 8) and (Equation 9), the following can be obtained:(10)$$\begin{array}{cc} \; & \min_{S}\sum_{v=1}^{V}\left\{\omega_{v}{∥C^{\left(v\right)}-S∥}_{\sigma }\right\}+\gamma \mathrm{Tr}\left(F^{T}LF\right)\\ \; & s.t.\;1^{T}s_{i}=1,s_{i,j}\geq 0,F^{T}F=I \end{array}$$

The objective function Equation (Equation 7) is finally transformed into Equation (Equation 10), and the consensus matrix *S* can be obtained only by solving it. By observing Equation (Equation 10), we can know that its second part is the objective function of spectral clustering, which ensures that *S* has *k* connected components; that is, the final community detection result can be obtained on *S* without executing other algorithms. Therefore, the consensus matrix *S* learned by the above method can complete the community detection of social networks of multi-view and obtain the community structure.

### 4.3. Algorithm Optimization

There are multiple unknown variables in the objective function Equation (Equation 10), so it will be very difficult to solve all variables at the same time. In order to obtain the optimal solution, we use the alternating iteration method to optimize the objective function. More specifically, we can choose to update one while keeping the others unchanged.

Step 1. Keep *F*, ω fixed and update *S*: When *F* and ω are fixed, using the property of Laplace matrix $\sum_{i,j}\frac{1}{2}{∥f_{i}-f_{j}∥}_{2}^{2}s_{i,j}=\mathrm{Tr}\left(F^{T}LF\right)$, then Equation (Equation 10) becomes:(11)$$\begin{array}{cc} \; & \min_{S}\sum_{v=1}^{V}\omega_{v}{∥C^{\left(v\right)}-S∥}_{\sigma }+\gamma \sum_{i,j=1}^{n}{∥f_{i}-f_{j}∥}_{2}^{2}s_{i,j}\\ \; & s.t.\;1^{T}s_{i}=1,s_{i}\geq 0 \end{array}$$

Define a matrix $E\in R^{n\times n}$, where $e_{i}\in R^{n\times 1}$ is the *i*-th column of *E*, and its *j*-th element is $e_{i,j}={∥f_{i}-f_{j}∥}_{2}^{2}$. Meanwhile, according to the research of Nie et al. [39], and the independence of each line in *S*, Equation (Equation 11) can be written in vector form:(12)$$\begin{array}{cc} \; & \min_{s_{i}}\sum_{v=1}^{V}\omega_{v}u_{i}^{\left(v\right)}{∥c_{i}^{\left(v\right)}-s_{i}∥}_{2}^{2}+\gamma s_{i}^{T}e_{i}\\ \; & s.t.\;1^{T}s_{i}=1,s_{i}\geq 0 \end{array}$$
where $s_{i}$ is the column vector composed of the *i*-th row element of *S*, and $c_{i}^{\left(v\right)}$ is the column vector composed of the *i*-th row element of the connected matrix $C^{\left(v\right)}$ of view *v* in the social network. $u_{i}^{\left(v\right)}$ can be calculated by:(13)$$\begin{array}{c} u_{i}^{\left(v\right)}=\left(1+\sigma \right)\frac{{∥c_{i}^{\left(v\right)}-s_{i}∥}_{2}+2\sigma }{2{\left({∥c_{i}^{\left(v\right)}-s_{i}∥}_{2}+\sigma \right)}^{2}} \end{array}$$

Equation (Equation 12) can be reduced to:(14)$$\begin{array}{cc} \; & \min_{s_{i}}\sum_{v=1}^{V}\frac{1}{2}\omega_{v}{u_{i}}^{\left(v\right)}s_{i}^{T}s_{i}-s_{i}^{T}\left(\sum_{v=1}^{V}\omega_{v}{u_{i}}^{\left(v\right)}c_{i}^{\left(v\right)}-\frac{\gamma }{2}e_{i}\right)\\ \; & s.t.\;1^{T}s_{i}=1,s_{i}\geq 0 \end{array}$$

Let $h_{i}=\sum_{v=1}^{V}\omega_{v}u_{i}^{\left(v\right)}$ and $p_{i}=\sum_{v=1}^{V}\omega_{v}u_{i}^{\left(v\right)}c_{i}^{\left(v\right)}-\frac{\gamma }{2}e_{i}$, Equation (Equation 14) can be simplified as:(15)$$\begin{array}{cc} \; & \min_{s_{i}}\frac{1}{2}h_{i}s_{i}^{T}s_{i}-s_{i}^{T}p_{i}\\ \; & s.t.\;1^{T}s_{i}=1,s_{i}\geq 0 \end{array}$$

Using the Lagrange multiplier method, we can obtain:(16)$$\begin{array}{c} \ell \left(s_{i},\eta ,\xi \right)=\frac{1}{2}h_{i}s_{i}^{T}s_{i}-s_{i}^{T}p_{i}-\eta \left(1^{T}s_{i}-1\right)-\xi^{T}s_{i} \end{array}$$

Here, η, ξ is the Lagrange multiplier that two constraints of Equation (Equation 14), η is a scalar, ξ is a vector. According to KKT conditions:

(17)∀j,hi,js^i,j − pi,j − η^ − ξ^j = 0(18)∀j, η^ ≥ 0(19)∀j, ξ^j ≥ 0(20)∀j,s^i,jξ^j = 0
where ${\hat{s}}_{i,j}$ is the optimal solution, $\hat{\eta }$ and ${\hat{\xi }}_{j}$ represents the corresponding Lagrange multiplier. Express Equation (17) as a vector with $h_{i}{\hat{s}}_{i}-p_{i}-\hat{\eta }1-\hat{\xi }=0$. Since $1^{T}s_{i}=1$, the following equation can be obtained.
(21)$$\begin{array}{c} \hat{\eta }=\frac{h_{i}-1^{T}p_{i}-1^{T}\hat{\xi }}{n} \end{array}$$

Therefore, the optimal solution ${\hat{s}}_{i}$ can be obtained and expressed as follows:(22)$$\begin{array}{c} {\hat{s}}_{i}=\frac{p_{i}}{h_{i}}+\frac{1}{n}+\frac{1^{T}p_{i}1}{nh_{i}}-\frac{1^{T}\hat{\xi }1}{nh_{i}}+\frac{\hat{\xi }}{h_{i}} \end{array}$$

Let $g=\frac{p_{i}}{h_{i}}+\frac{1}{n}+\frac{1^{T}p_{i}1}{nh_{i}}$ and ${\hat{\xi }}^{*}=\frac{1^{T}\hat{\xi }}{nh_{i}}$, Equation (Equation 22) can be written as ${\hat{s}}_{i}=g-{\hat{\xi }}^{*}1+\frac{\hat{\xi }}{h_{i}}$, and for any *j*, we have:(23)$$\begin{array}{c} {\hat{s}}_{i,j}=g_{j}-{\hat{\xi }}^{*}+\frac{{\hat{\xi }}_{j}}{h_{i,j}} \end{array}$$

According to Equation (18) to Equation (Equation 23), we have:(24)$$\begin{array}{c} {\hat{s}}_{i,j}=\max\left(g_{j}-{\hat{\xi }}^{*},0\right) \end{array}$$

It can be known by observing Equation (Equation 24), after ${\hat{\xi }}^{*}$ determination, the optimal solution ${\hat{s}}_{i,j}$ can also be determined. According to Equation (Equation 23), ${\hat{\xi }}_{j}=h_{i,j}\left({\hat{s}}_{i,j}-g_{j}+{\hat{\xi }}^{*}\right)$ can be deduced, and reuse Equation (18) to Equation (20), we have:(25)$$\begin{array}{c} {\hat{\xi }}_{j}=h_{i,j}\max\left({\hat{\xi }}^{*}-g_{j},0\right) \end{array}$$

Since ${\hat{\xi }}^{*}=\frac{1^{T}\hat{\xi }}{nh_{i}}$, according to Equation (Equation 25):(26)$$\begin{array}{c} {\hat{\xi }}^{*}=\frac{1}{n}\sum_{j=1}^{n}\max\left({\hat{\xi }}^{*}-g_{j},0\right) \end{array}$$

Define function $\xi^{*}$ as:(27)$$\begin{array}{c} f\left(\xi^{*}\right)=\frac{1}{n}\sum_{j=1}^{n}\max\left(\xi^{*}-g_{j},0\right)-\xi^{*} \end{array}$$

Therefore, we only need to know the root of $f\left({\hat{\xi }}^{*}\right)=0$ and solve ${\hat{\xi }}^{*}$. Since ${\hat{\xi }}^{*}\geq 0$ and $f\left({\hat{\xi }}^{*}\right)\leq 0$ is a piecewise linear convex function, the root of $f\left(\xi^{*}\right)=0$ can be solved by the Newton method, that is:(28)$$\begin{array}{c} \xi_{t+1}^{*}=\xi_{t}^{*}-\frac{f\left(\xi_{t}^{*}\right)}{f^{\prime }\left(\xi_{t}^{*}\right)} \end{array}$$

Step 2. Keep *F*, *S* fixed and update ω. When *F* and *S* are fixed, Equation (Equation 10) is equal to:(29)$$\begin{array}{cc} \; & \min_{S}\sum_{v=1}^{V}\omega_{v}{∥C^{\left(v\right)}-S∥}_{\sigma }\\ \; & s.t.\;1^{T}s_{i}=1,s_{i,j}\geq 0 \end{array}$$

At this time, we can solve the above equation to obtain $\omega_{v}$. According to the properties of the adaptive loss function, Equation (Equation 29) will be converted into:(30)$$\begin{array}{cc} \; & \min_{S}\sum_{v=1}^{V}w_{v}tr\left({\left(C^{\left(v\right)}-S\right)}^{T}U^{\left(v\right)}\left(C^{\left(v\right)}-S\right)\right)\\ \; & s.t.\;1^{T}s_{i}=1,s_{i,j}\geq 0 \end{array}$$
where $U^{\left(v\right)}$ is a diagonal matrix, and its *i*-th diagonal element is calculated by Equation (Equation 13). Then build the auxiliary function as follows:(31)$$\begin{array}{cc} \; & \min\sum_{v=1}^{V}\sqrt{\mathrm{tr}\left({\left(C^{\left(v\right)}-S\right)}^{T}U^{\left(v\right)}\left(C^{\left(v\right)}-S\right)\right)}\\ \; & s.t.\;1^{T}s_{i}=1,s_{i,j}\geq 0 \end{array}$$

Construct the Lagrange function $\min\sum_{i=1}^{V}\sqrt{t\left({\left(C^{\left(v\right)}-S\right)}^{T}U^{\left(v\right)}\left(C^{\left(v\right)}-S\right)\right)}+\Phi \left(\rho ,S\right)$ of Equation (Equation 31). Taking the partial derivative of *S* and making it equal to zero yields:(32)$$\begin{array}{c} \min\sum_{v=1}^{V}w_{v}\frac{\partial tr\left({\left(C^{\left(v\right)}-S\right)}^{T}U^{\left(v\right)}\left(C^{\left(v\right)}-S\right)\right)}{\partial S}+\frac{\partial \Phi (\rho ,S)}{\partial S}=0 \end{array}$$
where Φ(ρ,S) is the constraint term and ρ is the Lagrange multiplier, and: 
(33)$$\begin{array}{c} w_{v}=\frac{1}{2\sqrt{\mathrm{tr}{\left({\left(C^{\left(V\right)}-S\right)}^{T}U\left(C^{\left(V\right)}-S\right)\right)}^{2}}} \end{array}$$

It can be seen that making the partial derivative of the Lagrange function of Equation (Equation 31) with respect to *S* and make it equal to zero yields Equation (Equation 32), and substituting Equation (Equation 33) into Equation (Equation 32) is exactly equal to the partial derivative of the Lagrange function of Equation (Equation 31) with respect to *S* and makes it equal to zero. Thus, if ω is a constant, then solving Equation (Equation 30) is equivalent to solving Equation (Equation 31). At this point, the weight ω of each view is determined by Equation (Equation 33).

Step 3. Keep ω, *S* fixed and update *F*. When ω and *S* is fixed, it is equivalent to solving the following problem:(34)$$\begin{array}{cc} \; & \min_{F}\mathrm{Tr}\left(F^{T}LF\right)\\ \; & s.t.\;F^{T}F=I \end{array}$$

At this point, the optimal solution of *F* is composed of the eigenvectors corresponding to the smallest eigenvalues of the Laplace matrix *L* that ranked in the top *k*.

Note that the stopping condition for algorithm optimization is that the relative change in *S* is less than $10^{-3}$ or the number of iterations is greater than 150. The whole multi-view community detection process is shown in Algorithm 1.
**Algorithm 1** Multi-view community detection based on adaptive loss function (ALMV)
 **Input:** 
The association matrix $C^{\left(1\right)},C^{\left(2\right)},\ldots ,C^{\left(v\right)}$ of the social network of *V* views (Obtained by Equation (Equation 4); the number of communities *k*; initialization parameters γ, σ.**Output:** 
A consensus matrix *S* with *k* connected components.1:Initialize the weights ω=1/V of the connected matrix *C* for each view of the social network;2:Initialize the consensus graph matrix *S* (Through ω and *C*);3:Use Equation (Equation 34) to calculate the matrix *F*;4:**repeat**5:    Fix *S*, *F*, use Equation (Equation 30) to update ω;6:    Fix *F*, ω, use Equation Equation 24 to update *S*;7:    Fix ω, *S*, use Equation (Equation 34) to update *F*;8:**until** the relative change in *S* is less than $10^{-3}$ or the number of iterations is greater than 1509:**return** 
*S*


In order to understand and analyze social networks more easily, this paper introduces the Node2Vec graph embedding model [40] to visualize the results of community detection. It is a node vectorization model that obtains local information from truncated random wanderings, treating nodes as lexical items and wanderings as sentences to learn potential representations.

The process of community detection for social networks from multi-view has been described, and a summary of the overall process described above is shown in Algorithm 2.
**Algorithm 2** Multi-view community analysis method of social networks**Input:** Social network *G*; the number of nearest neighbors *m*; the number of communities *k*; the initialization parameters γ.**Output:** Visualization results of social network *G* containing *k* community structures.1:Filtering and splitting semantic information for *G*;2:Word frequency statistics of social network *G* to get the data matrix $X^{\left(1\right)}$;3:According to Equation (Equation 1), the TF-IDF value of each word in the *G* is calculated, and the data matrix $X^{\left(2\right)}$ of the keywords perspective is obtained using the method described in Section 2;4:Use the LDA topic model to obtain the topic distribution of *G* and get the data matrix of $X^{\left(3\right)}$ the topic perspective;5:**for**i=1; i<4; i++**do**6:    Input $X^{\left(i\right)}$;7:    Calculate the association between users in social networks using Equation (Equation 4);8:**end for**9:Input the above obtained $C^{\left(1\right)}$, $C^{\left(2\right)}$ and $C^{\left(3\right)}$ into Algorithm 1;10:Visualizing community detection results with Node2vec;11:**return** social networks of *k* community structure

## 5. Experiments

In this section, the experimental results of the proposed method on real social networks and public datasets are analyzed. The purpose of the experiment is to study the effectiveness of the proposed community detection method for social networks from multiple perspectives. All experiments in this paper use AMD ryzen7 5800h processor, 3.20 GHz, 16 GB RAM, and run in Python 3.8 and MATLAB r2018b development environment. Before further discussing the experimental process, the parameter setting is described here. In this paper, the default value of the number of nearest neighbors is m=22, with 1 as the initial value of parameter γ. The initial value will be automatically adjusted according to the number of iterations. When the connected component is less than the number of communities *k* during the construction of the consensus matrix, γ=γ×2. When it is greater than the number *k* of communities, γ=γ/2. Finally, according to research [39], the adaptive loss parameter is set to σ=0.1, in this case, record Algorithm 1 as ALMV-N1.5. When σ→0, ${∥X∥}_{\sigma }\to {∥X∥}_{2,1}$, Algorithm 1 is recorded as ALMV-N21. When σ→∞, ${∥X∥}_{\sigma }\to {∥X∥}_{F}$, Algorithm 1 is recorded as ALMV-NF.

### 5.1. Evaluation Index

To evaluate the performance of the proposed method, five metrics [41,42,43,44], accuracy (AC), normalized mutual information (NMI), adjusted Rand coefficient (AR), F-score, and modularity (*Q*), were used in the experiments.

Accuracy (AC). Given data $x_{i}$, let $g_{i}$ and $g_{i}^{\prime }$ represent the correct community and the predicted community, respectively. AC is defined as:
(35)$$\begin{array}{c} AC=\frac{\sum_{i=1}^{n}\delta \left(g_{i},{g^{\prime }}_{i}\right)}{n} \end{array}$$Here, *n* is δ(x,y), the total number of data. If x=y, then the function is equal to 1, otherwise it is equal to 0.Normalized Mutual Information (NMI). NMI represents the shared statistical information between the predicted and true categories. Given the correct category group $\Delta =\left\{g_{1},g_{2},\ldots ,g_{k}\right\}$ and the predicted category group $\Delta^{\prime }=\left\{{g^{\prime }}_{1},{g^{\prime }}_{2},\ldots ,{g^{\prime }}_{k}\right\}$ of the dataset *G*, let $p_{i}$ and $p^{\prime }$ denote the data points in categories Δ and $\Delta^{\prime }$, respectively, and $p_{st}$ denote the data points that are both in Δ and $\Delta^{\prime }$, the normalized mutual information of Δ and $\Delta^{\prime }$ is defined as:
(36)$$\begin{array}{c} \mathrm{NMI}=\frac{\sum_{s=1}^{c}\sum_{t=1}^{k}\log\left(\frac{np_{st}}{p_{s}{p^{\prime }}_{t}}\right)}{\sqrt{\left(\sum_{s=1}^{c}p_{s}\log\frac{p_{s}}{p}\right)\left(\sum_{t=1}^{k}{x^{\prime }}_{t}\log\frac{{p^{\prime }}_{t}}{p}\right)}} \end{array}$$Adjusted Rand coefficient (AR). AR is an optimized indicator based on the Rand coefficient (RI). Its formula is:
(37)$$\begin{array}{c} ARI=\frac{RI-E(RI)}{\max(RI)-E(RI)} \end{array}$$
where RI=(a+b)/(a+b+c+d) is the expected value of the Rand coefficient; *a* is a data point object that belongs to the same class in Δ, and also belongs to the same class in $\Delta^{\prime }$; *b* is a data point object that belongs to the same class in Δ and does not belong to the same class in $\Delta^{\prime }$; *c* is a data point object that does not belong to the same class in Δ, and belongs to the same class in $\Delta^{\prime }$; *d* is a data point object that does not belong to the same class in Δ, and also does not belong to the same class in $\Delta^{\prime }$.F-score. A comprehensive evaluation index that balances the impact of Accuracy and Recall. First, we introduce several basic concepts. TP (True positives): positive classes are judged as positive classes; FP (False positives): negative classes are judged as negative classes; FN (False negatives): positive classes are judged as negative classes; TN (True negatives): negative classes are judged as negative classes. F-score is defined as:
(38)$$\begin{array}{c} F=\frac{2\times Recall\times Accuracy}{Recall+Accuracy} \end{array}$$
where Accuracy=(TP+TN)/(TP+FN+FP+TN), Recall=TP/(TP+FN).Modularity (*Q*): Newman et al. [45] introduced modularity to assess the quality of community structure, which is defined as follows:
(39)$$\begin{array}{c} Q=\frac{1}{\left|O\right|}\sum_{i,j}\left(Sim_{i,j}-\frac{o_{i}o_{j}}{\left|O\right|}\right)\delta_{i,j} \end{array}$$
where $\left|O\right|$ is the sum of the degrees of all nodes in the network *G*; Sim is the Similarity matrix of the *G*; $o_{i}$ is the degree of node $p_{i}$; and $\delta_{i,j}$ is the Kronecker function, which is 1 if $p_{i}$ and $p_{j}$ are in the same community and 0 otherwise.

### 5.2. Experiment on Real Social Networks

The experiments in this section have two goals: (1) the social network reconstruction method proposed in this paper can be effectively applied to real networks, i.e., the data matrix of each perspective can be constructed based on different network attributes; (2) the performance of community detection from multiple perspectives on real social networks outperforms that of single-view community detection.

#### 5.2.1. Experiment Preparation

The data used in this section are composed of posts from Sina Weibo, including 10,176 Posts published by users from 1 March 2021 to 5 March 2021. To ensure the accuracy of the experiment, the data were cleaned (removing advertisements, repeated, short and other posts). Finally, 1584 posts were left as the final social network dataset.

We extracted features from three perspectives (word frequency, keywords, and topic) and constructed the data matrix for each perspective according to Equation (Equation 4). Our goal was to confirm that the community obtained by fusing the three features is better than the community in each single perspective. Therefore, we varied the number of features in each perspective separately to form multiple single-perspective networks, and selected the perspective with the highest *Q* value as the comparison perspective by performing community detection on each single-perspective network.

#### 5.2.2. Experimental Results

For the word frequency perspective, we set the word frequency to 2500, 5000, 7500, 10,000, 11,000, 12,500, 14,000, 15,000, and 20,000, respectively, and performed community detection on the formed network. The *Q* value of the generated communities are shown in Table 2, and the visual representation is shown in Figure 3.

From Table 2, we can see that the *Q* value was highest when the number of word frequency was equal to 12,500. Therefore, we chose the network reconstructed at this parameter as the word frequency perspective. Figure 3 shows that the three communities began to emerge when the word frequency was greater than 10,000, and the community characteristics were most evident when the word frequency reached 12,500 and 15,000.

For the keywords perspective, we set the number of keywords to 250, 500, 1000, 1500, 2000, 2500, 3000, 4000, and 5000, respectively, and performed community detection on the formed network. The *Q* value of the generated communities are shown in Table 3, and the visual representation is shown in Figure 4.

From Table 3, we can see that the *Q* value was highest when the number of keywords was equal to 3000. Therefore, we chose the network reconstructed at this parameter as the word frequency perspective. Figure 4 shows that the social network will be divided into three communities when the number of keywords is 2000 to 3000 under the same number of word frequency.

For the topic perspective, we set the number of topics from 0 to 100 with a span of 5, and performed community detection on the formed network. The *Q* value of the generated communities are shown in Figure 5, and the visual representation is shown in Figure 6.

From Figure 5a, we can see that the *Q* value decreases rapidly when the number of topics is greater than 55, and the *Q* value shows fluctuations when the number of topics is between 0 and 55. The *Q* value is highest when the number of topics is equal to 30. Therefore, we choose the network reconstructed at this parameter as the word frequency perspective. Figure 6 shows that none of the communities have clear boundaries when the number of topics ranges from 5 to 100, and the node distribution shifts from aggregation to dispersion as the number of topics increases.

Figure 5b depicts the *Q* values at different numbers of neighbors. When m>57, the *Q* value tends to remain steady, probably because the increase in the number of neighbors has little effect on the nodes with the greater similarity. Further, the weights of the edges between nodes with less similarity are small, which has little effect on the *Q* value. The selection of *m* is relatively wide, but it is optimal when m=22.

After the above process, we obtain three perspectives formed by word frequency, keywords, and topics, respectively. Next, we fuse the three perspectives using Algorithm 2 and execute the community detection method on the integrated network to verify the effectiveness of our method. We record the *Q* values of the communities and give a visualization in Figure 7.

In Figure 7, the *Q* value of the communities in word frequency perspective, keywords perspective, topic perspective, and multiview is 0.7441, 0.7165, 0.6637, and 0.7892, respectively. It can be seen that the communities in multiview have the highest *Q* values, which are 6.061%, 10.147%, and 18.909% higher than the word frequency, keywords, and topic perspectives, respectively. This indicates that integrating multiple perspectives for community detection can effectively improve community quality. In the visualization graph, the communities in multiview have clearer boundaries and almost no overlap.

### 5.3. Experiment on the Public Dataset

In this section, we conduct experiments on eight real-world datasets to verify that the ALMV algorithm proposed in this paper has excellent performance both in processing semantic datasets and in image datasets. It is also compared with the commonly used community detection methods to further evaluate the performance of ALMV. Table 4 lists the statistics of the corresponding characteristics of eight datasets, of which the first six datasets are semantic datasets and the last two are image datasets.

#### 5.3.1. Dataset

WebKB dataset (http://www.cs.cmu.edu/afs/cs.cmu.edu/project/theo-20/www/data/ accessed on 14 August 2022) (WebKB) [46]: This dataset consists of 203 pages in four categories collected by the Department of Computer Science at Cornell University. One page consists of three views: the page text content of the page, the anchor text on the link, and the text in the title.BBC dataset (http://mlg.ucd.ie/datasets/segment.html accessed on 14 August 2022) (BBC): This dataset comes from 250 BBC news sites, which correspond to five topics (business, entertainment, sports, science and technology, politics). It consists of 685 instances, each of which is divided into four parts, namely, four perspectives.BBC Sport dataset (http://mlg.ucd.ie/datasets/segment.html accessed on 14 August 2022) (BBCSports) [47]: This dataset is a documentation dataset consisting of sports news articles on five topics (track and field, football, tennis, rugby, and cricket) on the BBC Sports website from 2004 to 2005. Each article will extract two different types of features. It contains 685 samples with feature dimensions of 3183 and 333 from different perspectives.20 Newsgroups dataset (http://lig-membres.imag.fr/grimal/data.html accessed on 14 August 2022) (20NGs): The dataset consists of 20 different collections of newsgroup documents. It contains 500 different instances, each of which is preprocessed in three different ways.3Sources dataset (http://mlg.ucd.ie/datasets/3sources.html accessed on 14 August 2022) (3Sources): The dataset was collected from three online news organizations, the BBC, Reuters, and the Guardian, from February to April 2009. These three organizations reported 169 stories on one of six topics (entertainment, health, politics, business, sports, science, and technology).Wikipedia articles dataset (http://www.svcl.ucsd.edu/projects/crossmodal/ accessed on 14 August 2022) (Wikipedia) [48,49]: This dataset is a selection of files from a collection of Wikipedia featured articles. Each article has 2 perspectives and 10 categories; 693 instances are selected as experimental datasets.One-hundred plant species leaves dataset (https://archive.ics.uci.edu/ml/datasets/One-hundred+plant+species+leaves+data+set accessed on 14 August 2022) (100leaves) [50]: The dataset consists of 1600 samples from three perspectives, each of which is one of 100 species.Handwritten digit 2 source dataset (https://cs.nyu.edu/~roweis/data.html accessed on 14 August 2022) (HW2sources): The dataset was collected from 2000 samples from two sources: MNIST handwritten digits (0–9) and USPS handwritten digits (0–9).

#### 5.3.2. Baseline Method

To verify the performance of the methods presented in this paper, performance comparisons will be made between the following methods:Normalized cut (Ncut) [14]: Ncut is a typical graphics-based method, which is used in each perspective of each dataset to select the best performance perspective as the result. The parameters in the algorithm are set according to the author’s recommendations.Fast unfolding algorithm (Louvain) [16]: Louvain is a modularity-based community detection algorithm that discovers hierarchical community structures with the objective of maximizing the modularity of the entire graph’s attribute structure. In this paper, we construct the weight matrix by Gaussian kernel function and run the algorithm in a recursive manner.Clustering with Adaptive Neighbors (CAN) [51]: CAN is an algorithm that learns both data similarity matrix and clustering structure. It assigns an adaptive and optimal neighbor to each data point based on the local distance to learn the data similarity matrix. The number of iterations and parameters of this algorithm run in this paper are the default values set by the author.Smooth Representation (SMR) [52]: This method deeply analyzes the grouping effect of representation-based methods, and then sub-spatial clustering is performed by the grouping effect. When experimenting on datasets with this method, the parameters are set to α=20 and knn=4.Multi-View Deep Matrix Factorization (DMF) [53]: DMF can discover hidden hierarchical geometry structures and have better performance in clustering and classification. In this paper, the model is set into two layers, with the first layer having 50 implicit attributes.Co-regularized Spectral Clustering (CRSC) [24]: This method achieves multi-view clustering by co-regularizing the clustering hypothesis, which is a typical multi-view clustering method based on spectral clustering and kernel learning. It uses the default parameters set by the author.Multi-view Clustering with Graph Learning (MVGL) [54]: This is a multi-view clustering method based on graphics learning, which learns initial diagrams from data points of different views and further optimizes the initial diagrams using rank constraints of Laplace matrices.We set the number of neighbors to the default value of 10 for our experiment.Proximity-based Multi-View NMF (PMVNMF) [55]: It exploits the local and global structure of the data space to deal with sparsity in real multimedia (text and image) data and by transferring probability matrices as first-order and second-order approximation matrices to reveal their respective underlying local and global geometric structures. This method uses the default parameters set by the author.

Among the eight baseline methods mentioned above, the top four are algorithms that work on a single view, and the last four are multi-view clustering methods. Ncut is a widely used algorithm today and can be used for community detection and clustering; Louvain is a module-based community detection method; CAN and SMR are node-based community detection methods; DMF and CRSC are multi-view spectral clustering methods based on the k-NN algorithm and kernel learning, respectively; MVGL is a method based on multi-view clustering; PMVNMF is a new multi-view clustering method with better performance.

#### 5.3.3. Parameter Analysis

In this section, we verify the performance of our adaptive loss function (marked as ALMV-N1.5). Figure 8 shows the performance of ALMV algorithm on ACC, NMI, and *Q* value with σ=0.1, σ→0.1, and σ→∞. It can be seen that the performance of the multi-view community detection method depends on the type of loss function. AlMV-N21, which constructs the loss function with l21-norm, is significantly lower than ALMV-N1.5 in ACC and NMI, and the *Q* value is only slightly better than ALMV-N1.5 in WebKB and 3sources datasets. The performance of ALMV-NF, which constructs a loss function with F-norm is inferior to ALMV-N1.5 in ACC, NMI, and *Q* values. This shows that the performance of our adaptive loss function is better than the conventional loss function, which provides a new idea for the loss function construction in other fields.

#### 5.3.4. Experimental Result on Public Dataset

In this section, we compare the ALMV algorithm with eight baseline methods described in Section 5.3.2. We use eight real-world networks described in Section 5.3.1 as the experiment data, and the results are shown in Table 5 and Figure 9.

The AC, NMI, AR, and F-score of each algorithm are given in Table 5. The best results for each dataset experiment are highlighted in bold. We can see that the method proposed in this paper clearly outperforms all the baseline methods, showing the best performance on all datasets except for the AR and F-score on the 100leaves dataset, which are slightly lower than MVGL. The 100Leaves dataset contains 100 clusters, which makes ALMV vulnerable to the interference of information from other nodes in the same layer in capturing the features of each perspective, resulting in a degradation of the final community detection performance. Compared with the graph-based method MVGL, ALMV has a significant performance advantage over other datasets. The adaptive loss function used in ALMV can take advantage of the robustness of the l21-norm and F-norm, and automatically weighted the network when fusing it, effectively capturing the core information of each perspective.

Figure 9 shows the *Q* values of each algorithm on the eight datasets. Overall, the ALMV algorithm shows a strong competitive performance. ALMV’s performance is more stable and does not fluctuate as much with data changes. In the WebKB dataset, for example, the *Q* value of the ALMV algorithm is slightly lower than that of SMR and NCUT. NCUT, on the other hand, is unable to perform the community detection task on data with complicated links, such as Wikipedia and 100Leaves. Similarly, SMR also shows extremely low community detection performance on BBC, BBCSport, and NGs. The *Q* values of the ALMV algorithm are closer to those of MVGL and CAN on the 100Leaves dataset; however, MVGL and CAN are clearly more influenced by the data type and consequently exhibit more fluctuations in the results across datasets. Although the PMVNMF and Louvain algorithms are stable, their overall *Q* values are lower than those of the ALMV method.

In addition to providing the *Q* value, AC, AR, F-score, and NMI score of the algorithm on each dataset, we also give the visual presentation and matrix diagram of the algorithm on the 20NGs and 100Leaves datasets. As shown in Figure 10, each color in the figure represents a community. The visual graph of the multi-view community detection is represented by the term multiview. The visual graphs under the word frequency, keyword, and topic perspectives are denoted by the terms view1, view2, and view3, respectively. The community boundaries in view1 and view2 are much less obvious than in multiview, and the communities overlap considerably. ALMV gradually increases the weight of effective perspectives while decreasing the weight of less useful perspectives during the consensus matrix learning process, weakening the influence of invalid information on the final outcomes and improving ALMV’s community detection performance.

Figure 11 is the matrix diagram of the community detection results on the 20NGs and 100leaves datasets. We can observe that both single and multiple views can identify the number of connected components in the matrix graph, but the effect of multiple views is significantly improved compared to single views. For example, the contours of the connected components in Figure 11d are very fuzzy, the cohesiveness of the community is weak. Similarly, in Figure 11f–h, we can see that there are more outliers around the principal components, a situation that is more serious for a network with more clusters such as 100Leaves, which directly reduces the clustering coefficient of the network. ALMV will reduce the weight of such perspective during the consensus matrix learning process, thus improving the community detection performance.

## 6. Conclusions

This paper proposes a multi-view fusion method for semantic social networks based on adaptive loss function for community detection. We extract the text features of semantic social networks from three views: word frequency, keywords, and topics, and propose a new similarity calculation method based on sparse representation to reconstruct the network for each view. Combining the advantages of L21-norm and F-norm, we use the adaptive loss function to automatically weight the correlation matrix of each view. We embed the spectral clustering process in the objective optimization function, which enables the algorithm to output community structure while performing matrix fusion.

We compare the proposed method to seven representative algorithms on eight datasets. We discovered that: (1) when considering only one single view of the network, the modularity of community structure is low, and the visual graphics are unreasonable; (2) when considering multiple views of the network, the modularity of community structure is high, and the community has obvious boundaries in the visual graphics; (3) the method proposed in this paper can effectively reduce the impact of less important views on matrix fusion, which enables the community detection algorithm to achieve better performance in terms of modularity, accuracy, and F-score.

Dynamic properties are common in real semantic social networks. In future work, we will investigate how to design efficient adaptive algorithms to calibrate the existing community structure during network evolution to achieve online community detection.

## Figures and Tables

**Figure 1 entropy-24-01141-f001:**
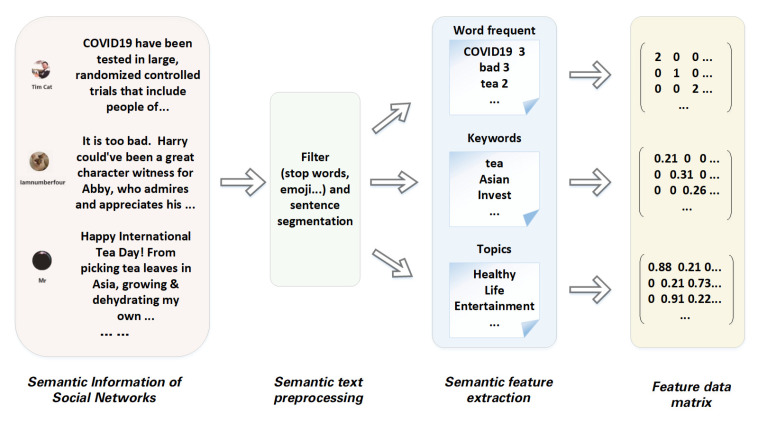
The process of multi-view feature representation of social networks.

**Figure 2 entropy-24-01141-f002:**
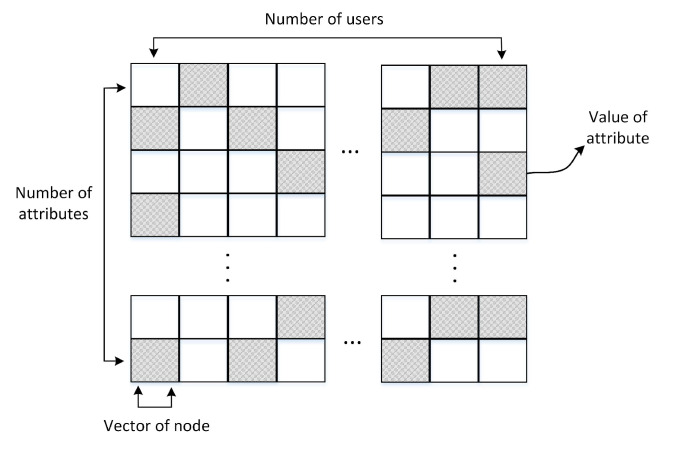
The data storage matrix for social networks.

**Figure 3 entropy-24-01141-f003:**
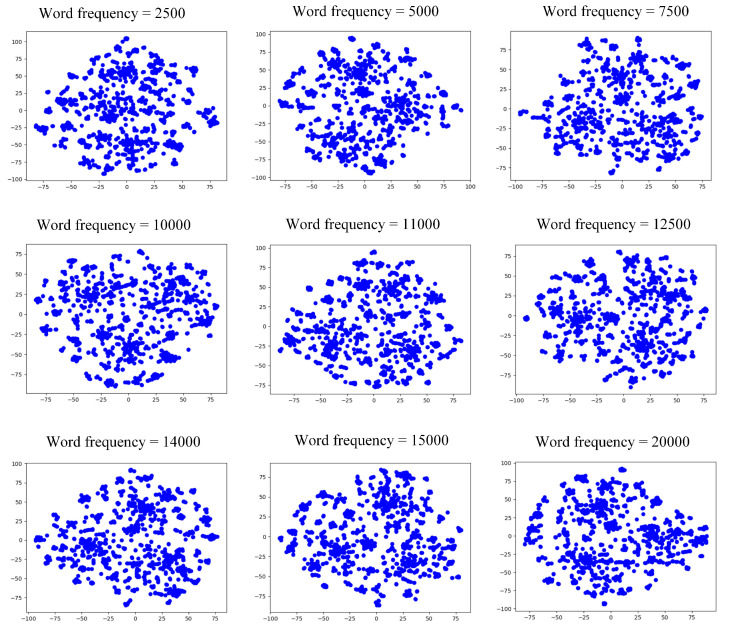
Community structure from the view of word frequency of microblog dataset.

**Figure 4 entropy-24-01141-f004:**
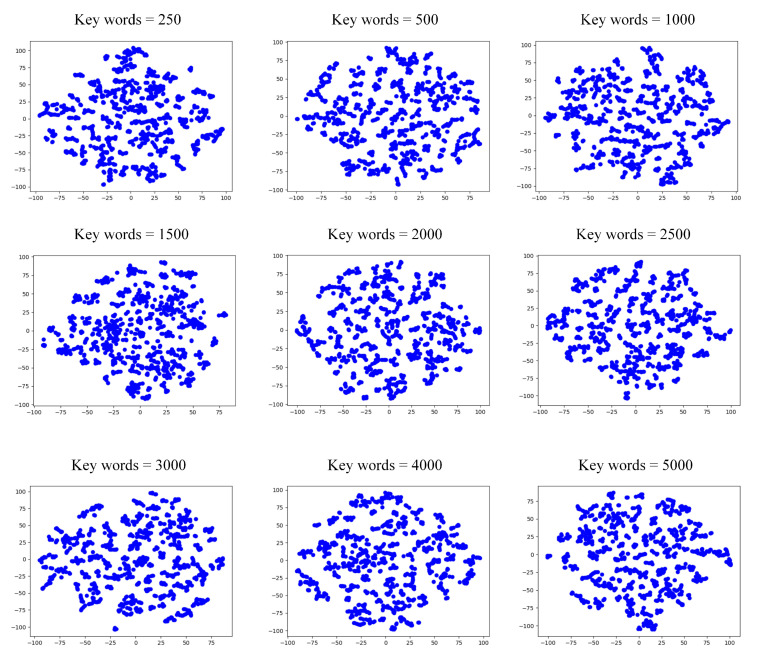
Community structure from the view of keywords of the microblog dataset.

**Figure 5 entropy-24-01141-f005:**
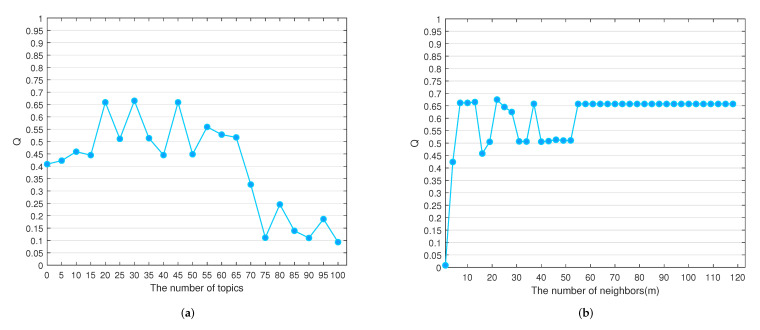
The *Q* value with different number of neighbors and topics. (**a**) Modularity *Q* varies with the number of topics (**b**) Modularity *Q* varies with the number of neighbors.

**Figure 6 entropy-24-01141-f006:**
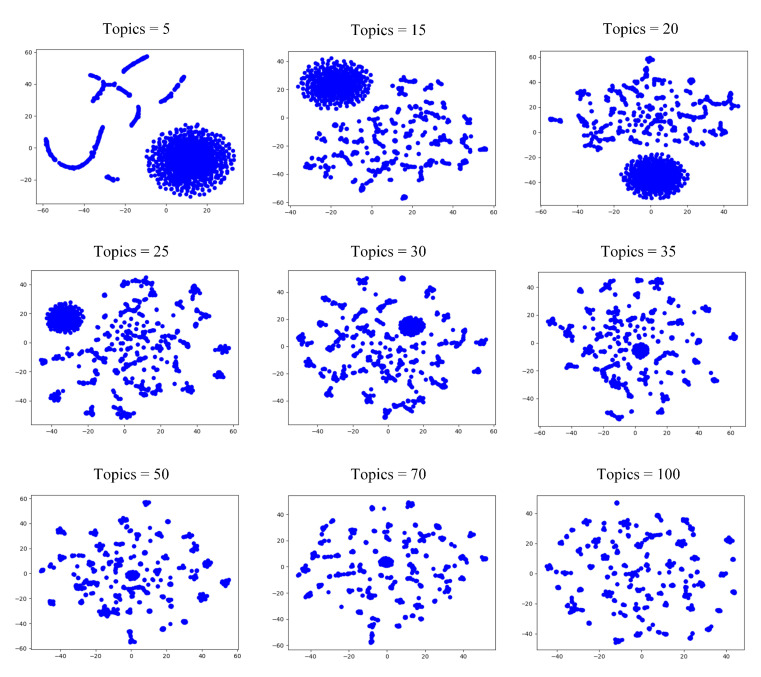
Community structure from the view of topic of microblog dataset.

**Figure 7 entropy-24-01141-f007:**
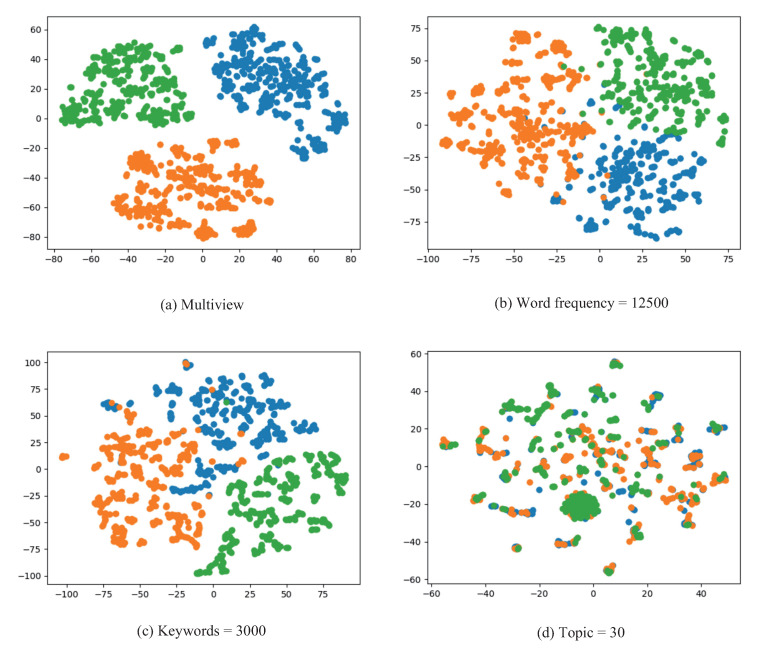
Result of single-view and multi-view community detection of microblog datasets with word frequency = 12,500, keyword = 3000, and topic = 30.

**Figure 8 entropy-24-01141-f008:**
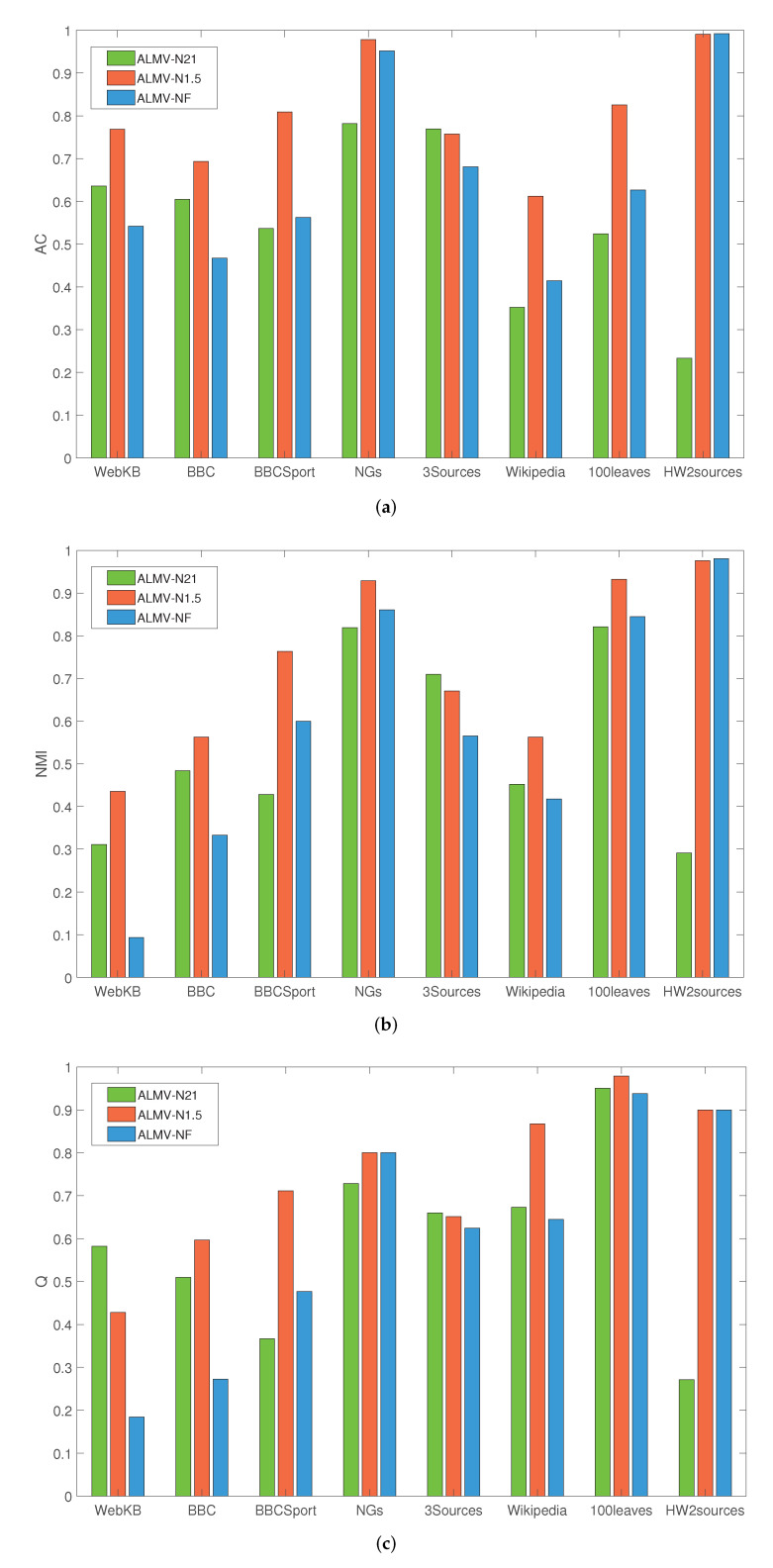
Performance ALMV with different parameter σ. (**a**) Accuracy; (**b**) Normalized Mutual Information; (**c**) Modularity.

**Figure 9 entropy-24-01141-f009:**
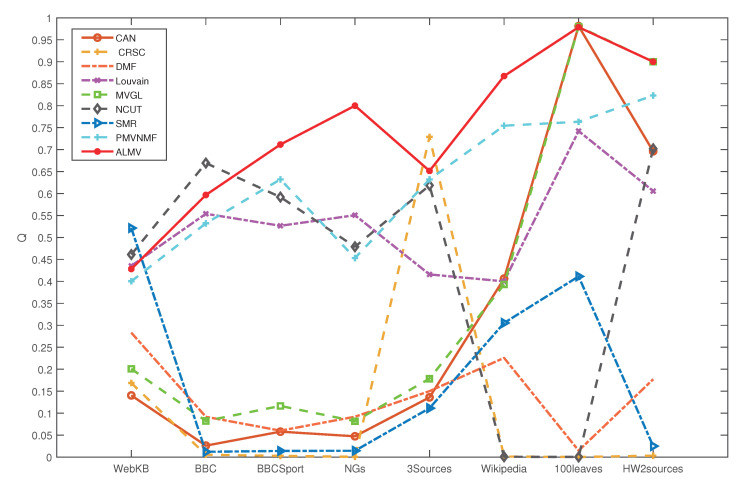
The *Q* value of the nine algorithms on eight datasets.

**Figure 10 entropy-24-01141-f010:**
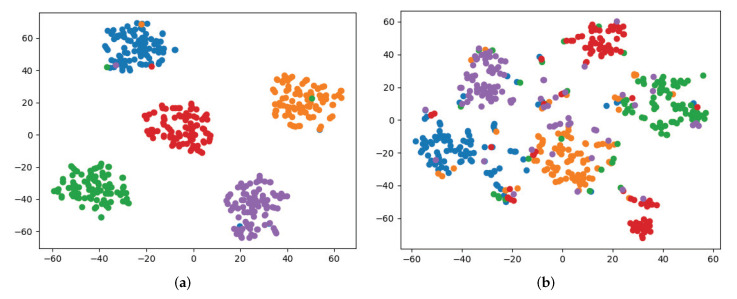
The results of running ALMV algorithm on 20NGs and 100leaves datasets. (**a**) 20NGs-multiview; (**b**) 20NGs-view1; (**c**) 20NGs-view2; (**d**) 20NGs-view3; (**e**) 100leaves-multiview; (**f**) 100leaves-view1; (**g**) 100leaves-view2; (**h**) 100leaves-view3.

**Figure 11 entropy-24-01141-f011:**
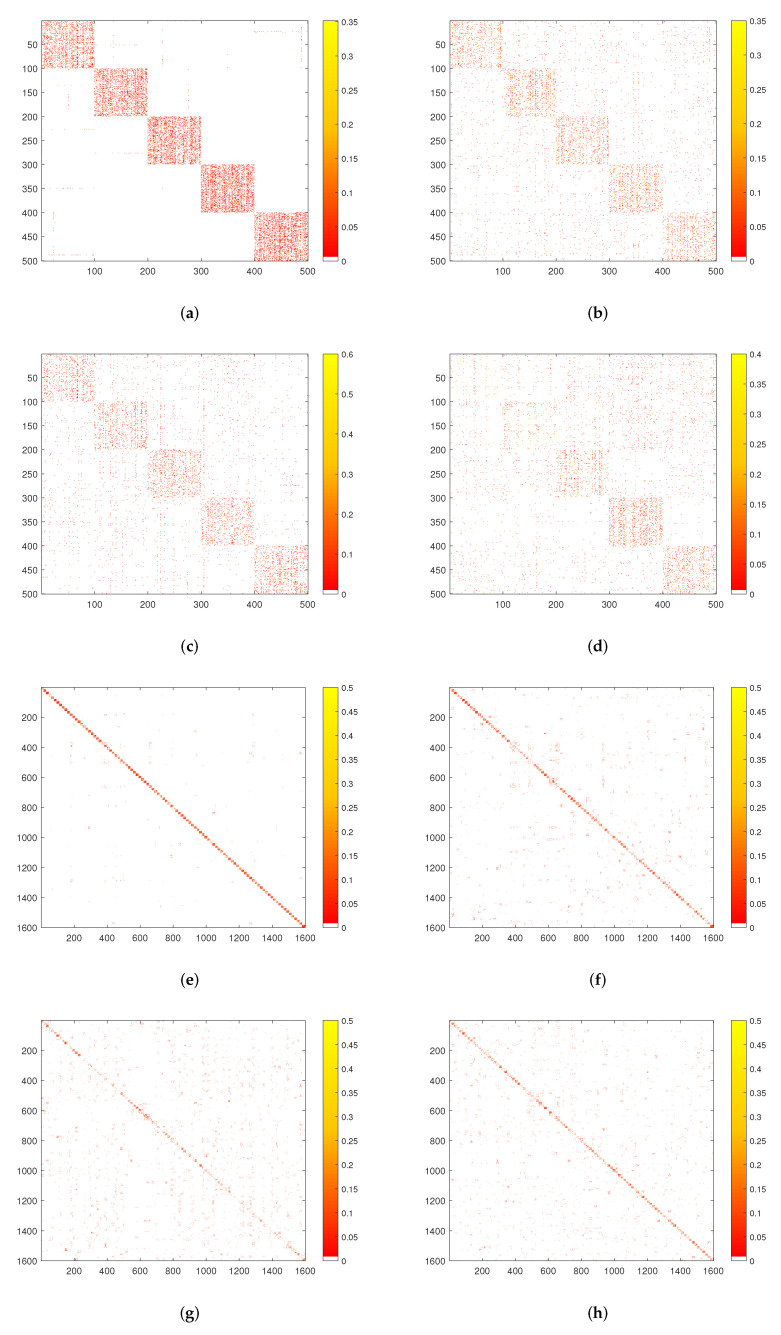
The matrix diagram of running ALMV algorithm on 20NGs and 100leaves datasets. (**a**) 20NGS-multiview; (**b**) 20NGs-view1; (**c**) 20NGs-view2; (**d**) 20NGs-view3; (**e**) 100leaves-multiview; (**f**) 100leaves-view1; (**g**) 100leaves-view2; (**h**) 100leaves-view3.

**Table 1 entropy-24-01141-t001:** Description of notations.

Notation	Description
*G*	Social network
*I*	Identity matrix
1	Column vector with all 1 element
*C*	Connected matrix
*S*	Consensus matrix
ω	The weight of single-view
*k*	The number of communities
*V*	The number of analysis views
Tr(X)	Trace of matrix *X*
${∥X∥}_{F}$	Frobenius norm of matrix *X*
${∥X∥}_{i}$	i-norm of matrix *X*
${∥x∥}_{i}$	i-norm of matrix *x*

**Table 2 entropy-24-01141-t002:** Modularity of the network reconstructed by word frequency.

Word Frequency	Modularity *Q*
2500	0.5342
5000	0.6663
7500	0.6701
10,000	0.7024
11,000	0.7392
12,500	0.7441
14,000	0.7172
15,000	0.7263
20,000	0.6972

**Table 3 entropy-24-01141-t003:** Modularity of the network reconstructed by keywords.

Keywords	Modularity *Q*
250	0.6195
500	0.6421
1000	0.6376
1500	0.6710
2000	0.7001
2500	0.6986
3000	0.7165
4000	0.6781
5000	0.6734

**Table 4 entropy-24-01141-t004:** Description of multi-view datasets.

Dataset	Categories	View	Samples	Features
WebKB	4	3	203	1703/230/230
BBC	5	4	685	4659/4633/4665/4684
BBCSport	5	2	544	3183/3203
20NGs	5	3	500	2000/2000/2000
3Sources	6	3	169	3560/3631/3068
Wikipedia	10	2	693	128/10
100leaves	100	3	1600	64/64/64
HW2sources	10	2	2000	784/256

**Table 5 entropy-24-01141-t005:** Performance comparison of the eight algorithms on eight datasets.

Dataset	Index	Ncut	Louvain	CAN	SMR	DMF	CRSC	MVGL	PMVNMF	ALMV
WebKB	AC(%)	72.41	48.77	56.16	65.52	64.04	70.44	30.18	71.23	76.85
	NMI(%)	28.73	38.23	9.24	34.51	25.29	27.18	10.90	39.22	43.51
	AR(%)	36.03	35.82	5.08	37.49	31.46	34.49	3.61	31.56	43.97
	F-score(%)	64.04	54.12	56.29	62.47	57.19	61.54	33.92	69.23	70.04
BBC	AC(%)	32.56	21.31	34.16	48.91	28.38	33.14	34.74	48.53	69.34
	NMI(%)	2.66	41.41	4.40	36.57	7.42	1.88	6.40	40.56	56.28
	AR(%)	0.07	10.07	0.59	17.10	3.65	0.23	0.15	44.87	47.89
	F-score(%)	37.70	13.71	38.07	40.27	25.39	37.93	37.46	50.63	63.33
BBCSports	AC(%)	35.66	20.04	36.58	71.51	32.54	35.85	40.07	73.52	80.88
	NMI(%)	1.29	48.56	4.36	56.06	6.24	1.81	14.24	63.21	76.35
	AR(%)	0.23	10.69	0.48	46.24	3.63	0.26	4.02	52.90	72.78
	F-score(%)	38.36	14.09	38.48	61.55	26.02	38.43	40.08	66.71	79.88
20NGs	AC(%)	21.20	25.60	23.00	46.60	36.80	21.80	22.80	34.64	97.80
	NMI(%)	2.11	47.40	6.98	32.61	10.97	2.96	77.51	47.65	92.87
	AR(%)	0	15.67	0.31	17.16	7.95	0.06	0.23	13.98	94.57
	F-score(%)	38.36	14.09	38.48	61.55	26.02	38.43	40.08	37.78	95.65
3Sources	AC(%)	33.14	49.11	35.50	49.11	37.22	31.36	22.80	56.82	75.74
	NMI(%)	4.20	62.55	10.74	41.62	23.78	7.92	7.51	46.95	67.05
	AR(%)	−0.21	39.91	0.04	22.84	10.4	3.55	0.23	40.31	53.70
	F-score(%)	29.12	47.84	36.36	42.77	30.86	28.34	32.78	55.12	66.55
Wikipedia	AC(%)	52.81	19.63	53.82	58.30	50.79	51.37	24.39	43.34	61.18
	NMI(%)	50.19	5.52	55.57	55.23	51.76	40.20	20.33	52.58	56.25
	AR(%)	35.32	1.73	31.46	41.45	33.98	33.42	0.73	36.60	45.27
	F-score(%)	43.09	14.78	40.88	48.42	41.57	40.50	19.10	44.12	51.80
100leaves	AC(%)	47.63	57.19	63.69	33.75	23.87	75.06	76.56	71.32	82.56
	NMI(%)	72.36	81.59	83.62	65.86	54.68	90.39	89.29	83.75	93.25
	AR(%)	31.15	42.18	42.94	20.99	9.39	69.41	50.62	57.12	54.58
	F-score(%)	31.97	42.89	43.65	22.12	10.31	69.73	51.25	51.76	55.17
HW2sources	AC(%)	11.90	53.40	48.25	46.15	40.71	68.35	98.45	96.32	99.05
	NMI(%)	1.33	57.33	60.45	43.27	36.78	61.01	96.20	95.32	97.61
	AR(%)	0	40.46	30.88	29.31	22.68	53.15	96.59	92.61	97.90
	F-score(%)	16.01	46.24	41.25	36.93	30.50	57.86	96.93	96.73	98.11

## Data Availability

The publicly available datasets analyzed for this study can be found in http://www.cs.cmu.edu/~WebKB/ (accessed on 14 August 2022). Further inquiries can be directed to the corresponding author.

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
