# Peer review of "Community Detection in Semantic Networks: A Multi-View Approach"

_entropy, 2022, doi:10.3390/e24081141_

Round 1

Reviewer 1 Report

The paper presents a new method for community identification in semantic networks, which is proposed in multi-view aspect. The method is well described and verified through usage of eight different datasets. The results are discussed and the future work is planned. 

The authors should check carefully the manuscript for grammar errors and typos, which are in the following form:

- topic Zd,n depends on the topic... - the sentence has to start with capital letter;

- Zd,n[33].- interval between the parameter and the reference;

- T(the method is given in Section 5) - space between T and the bracket;

- After obtaining the data matrix of each angle, the connected matrix is obtained by calculating the - the word obtain should be replaced by other similar word when it is used twice in one sentence; 

- ... by the user is small) - missing point for the end of the sentence;

- ... among α Is a sparse factor - is should be with small letter;

- this matrix is called the consensus matrix S ∈ Rn×n, - the end of the sentence has to end with point;

- among σ Make an adap.. - make should be with small letter m;

- order λi(L) Represents the i-th - small and capital letters;

The pictures in Figure 9 should be bigger for better readability.

The pictures in Figure 10 should be with light background and bigger.

A new section Discussion could be created after section 5 Experiments as part of explanations from section 5 could be moved here and also the pictures from the last section Conclusions.

Author Response

Thank you for your comments on our manuscript. Please see the attachment for the responses to the review comments.

Reviewer 2 Report

Community Detection in Semantic Networks: A Multi-View Approach

The considered manuscript proposes a method for community detection in semantic social networks. The paper does not develop or use the concept of entropy as such, but it studies networks as complex systems. Hence, I find it reasonably relevant to the scope of the Entropy journal.
My overall impression of the paper is positive: it has a comprehensive structure and good technical quality, details the theoretical/mathematical apparatus, and performs extensive (though not so intensive) evaluation. The two main problems that I believe need improvement are related to overall novelty/contributions and to rigor in sub-section 5.2. After these are corrected, I think the manuscript can be accepted for the publication in Entropy.

First, in my opinion, the novelty of the work described in the paper and the results are not justified enough. In essence, the proposed method is composed from several very well known components (Frequency Analysis, LDA, etc.). So, the authors need to better review the field and the up-to-date approaches, including multi-view ones. Currently, the baselines that the authors use give the impression of a "straw man", which is too easy to beat. I believe at least a couple of advanced multi-view methods should be employed. Contributions of the paper (63-73) need to be re-formulated, to reflect novel findings and take-aways. Currently, they rather tell about details of the algorithm, whose value for an external researcher are not apparent.
    Second, the analysis of the parameters in 5.2.1 is much weaker in rigor than the rest of the paper. The authors repeatedly propose to just "look and see" in the graphs, using expressions "results are obviously better" and even "results are the most ideal". I believe the whole approach needs to be re-considered and the subsection needs to be re-written to be more mathematically and scientifically justified.

Misc
* Some figures (e.g.,Fig. 1) should be made larger, currently the texts are largely unreadable.
* Sections should not have only a single subsection (e.g., 5.2., 5.2.1., then 5.3 without any 5.2.2.).
* 64: "We extracting semantic features..." -> "We extract semantic features..."

Author Response

(The authors gave the same response as above.)

Round 2

Reviewer 2 Report

I have read the authors' replies to both reviewers, as well as the updated version of the manuscript. I'm happy to see that the authors have convincingly addressed all the recommendations. So, I recommend accepting the paper and commend the authors for their work.

Some minor issues:

I recommend to re-consider Table 2. Currently it gives a reader the impression that the two parameters, Word frequency and Keywords, are somehow related. However, from what I understand from the paper, each parameter is are altered independently in 5.2.2. (until Fig. 7a). If so, Table 2 should be divided into two separate ones.

The text in Fig. 8 is too small.

Author Response

Dear Reviewer :

Thank you for your comments on our manuscript. The comments are very helpful for revising and improving our manuscript. According to your detailed suggestions, we have made a careful revision of our manuscript.

1. We have divided Table 2 into two separate tables.

2. We have increased the font size of the text in Figure 8.

Thank you again for your comments.